# Newly Emerging Airborne Pollutants: Current Knowledge of Health Impact of Micro and Nanoplastics

**DOI:** 10.3390/ijerph18062997

**Published:** 2021-03-15

**Authors:** Alessio Facciolà, Giuseppa Visalli, Marianna Pruiti Ciarello, Angela Di Pietro

**Affiliations:** Department of Biomedical and Dental Sciences and Morphofunctional Imaging, University of Messina, Via C. Valeria, I-98100 Messina, Italy; afacciola@unime.it (A.F.); gvisalli@unime.it (G.V.); marianna.pruiti@gmail.com (M.P.C.)

**Keywords:** microplastics (MPs), nanoplastics (NPs), indoor and outdoor exposure, toxicity of airborne MP-NPs, oxidative stress, inflammation

## Abstract

Plastics are ubiquitous persistent pollutants, forming the most representative material of the Anthropocene. In the environment, they undergo wear and tear (i.e., mechanical fragmentation, and slow photo and thermo-oxidative degradation) forming secondary microplastics (MPs). Further fragmentation of primary and secondary MPs results in nanoplastics (NPs). To assess potential health damage due to human exposure to airborne MPs and NPs, we summarize the evidence collected to date that, however, has almost completely focused on monitoring and the effects of airborne MPs. Only in vivo and in vitro studies have assessed the toxicity of NPs, and a standardized method for their analysis in environmental matrices is still missing. The main sources of indoor and outdoor exposure to these pollutants include synthetic textile fibers, rubber tires, upholstery and household furniture, and landfills. Although both MPs and NPs can reach the alveolar surface, the latter can pass into the bloodstream, overcoming the pulmonary epithelial barrier. Despite the low reactivity, the number of surface area atoms per unit mass is high in MPs and NPs, greatly enhancing the surface area for chemical reactions with bodily fluids and tissue in direct contact. This is proven in polyvinyl chloride (PVC) and flock workers, who are prone to persistent inflammatory stimulation, leading to pulmonary fibrosis or even carcinogenesis.

## 1. Introduction

Plastics are synthetic polymeric organic materials largely used for their remarkable versatility and cost-effectiveness, with an annual worldwide production of 359 million tons. Asia is the major producer, with 183 million tons (51%) produced in 2018. In particular, China alone is responsible for 30% of the total, accounting for 107.7 million tons of plastic materials, followed by the Americas, with 64.6 million tons (18%), as reported by North American Free Trade Agreement (NAFTA), and the European Union (EU) with 61.8 million tons (17%) (Plastics Europe, 2019) [1,2]. Based on specific characteristics, two different groups known as “thermoplastics” and “thermosets” can be distinguished in this composite family. The first group is made up of several polymeric materials that can be repeatedly melted when heated and hardened when cooled, and include polyethylene (PE), polypropylene (PP), polycarbonate (PC), polyvinyl chloride (PVC), polyethylene terephthalate (PET), and polystyrene (PS). In the second group, the polymerization process is irreversible, and once the three-dimensional network is formed, the materials cannot be re-melted to form new products [3]. 

Due to inappropriate waste management of the produced plastics, they are ubiquitous persistent pollutants, forming the most representative material of the Anthropocene. It is estimated that more than 250,000 tons of plastic litter is floating in the oceans, having accumulated over time [4]. In the environment, plastic objects undergo wear and tear (i.e., mechanical fragmentation, and slow photo and thermo-oxidative degradation) and, to a minor degree, biodegradation. These processes deteriorate material integrity with the formation of pieces smaller than 5 mm, called microplastics (MPs) [5], which are known as secondary MPs. However, plastic particles of this size are also intentionally produced to be used especially in cosmetic products (e.g., exfoliants or toothpaste) or by industries (e.g., air blasting). These are called primary microplastics [6]. 

MPs, from which nanoplastics (NPs) are derived following further fragmentation, have been found in many environmental sites globally [7,8,9]. Due to their environmental persistence and continuous release, they have already been detected in seawater, at concentrations up to 102,000 particles/m^3^ [10], and in sediment [11,12], soil [13], air [14,15,16], and foodstuffs, such as beer, sea salt, and bottled and tap water [17,18]. Due to biaccumulation, MPs have also been detected in several organisms along the trophic scale [19,20], in which adverse effects have been verified and highlighted in laboratory experiments [21,22]. It has been shown that the amounts of released MPs are correlated with larger population densities, and the distribution is influenced by environmental factors, such as currents and winds, and by the particle concentrations [23]. These factors are often responsible for the movement of MPs between the different environmental compartments [24]. The environmental transport and distribution of plastic debris depend on factors linked to the intrinsic properties of MPs, i.e., shape, size, and density. In particular, regarding the shape, MPs have been divided into fragments (with a length to width ratio ≤3), microfibers (length to width ratio >3), films, foam, and beads [25]. Despite fragments being considered the most abundant form, fibers appear to be far more represented and account for 67% of all atmospheric MPs in Shanghai [26] and 92% in London [27].

According to their size, MPs have been divided into medium-sized MPs (1.01–5 mm) and small MPs (<1.00 mm) [4], and NPs have at least one size within <100 nm. NPs can more easily penetrate into tissues and cells and can accumulate in organs causing alterations of physiological processes [28]. Moreover, due to “quantum confinement effects”, their nanoscale size confers different chemical-physical characteristics in comparison to the same material in the macro- or microscale [29]. In nanoparticles, the number of surface area atoms per unit mass is increased by several orders of magnitude, greatly enhancing the surface area for chemical reactions with bodily fluids and tissue in direct contact with particle surface [30]. The density of plastic particles ranges from 0.85 to 1.41 g/cm^3^ [31]. In addition to determining a different distribution in water bodies (floating on the water or settling on the seabed), a lower density of MPs favors a long stay in the air.

Several studies have been conducted to investigate human exposure to MP-NPs and the possible effects on human health [32,33,34,35]. To date, most of these studies have focused on the gastrointestinal system because exposure via the ingestion of contaminated food and water was considered to be the predominant route in humans, causing the intestinal uptake of MPs [36,37,38,39,40,41]. Only a few reviews were focused on human exposure by inhalation [33,42,43,44]. However, Catarino et al. [38] showed how the ingestion of synthetic fibers from the consumption of mussels is lower than that following the inhalation of household dust during the same meal. Inhalation plays a pivotal role in human exposure to micro- and nanoparticles, such as airborne particulate matter and both metal- and carbon-based engineered nanoparticles. The pathogenic mechanism of these airborne particles has been proven in several in vitro and in vivo studies [45,46,47,48,49]. Through inhalation, any type of smaller microparticle can reach the alveolar surface and nanosized particles can easily pass into the bloodstream, overcoming the pulmonary epithelial barrier (Figure 1). Their distribution causes damage to different body regions, including the central nervous system (CNS) [50,51].

To assess potential health damage due to human exposure to airborne micro- and nanoplastics, in this descriptive review we summarize the evidence collected to date, which has, however, almost completely focused on monitoring and on the effects of airborne microplastics. Only a limited number of in vitro studies have assessed the toxicity of nanoplastics [52] and a standardized method for their analysis in environmental matrices is still missing.

## 2. Sources and Concentration of Airborne MP-NPs

Human exposure to airborne MPs depends on the wide presence and distribution of their sources, including the wear-and-tear of synthetic textile fibers and rubber tires, meaning that they are abundantly present in city dust [53]. Other secondary sources could be represented by upholstery and household furniture [15,54,55], buildings, incinerators, landfills [14], industrial emissions, wear-and-tear of vehicle components [15], and materials used in agriculture (e.g., PS peat and sewage sludge used as fertilizer) [55]. Because synthetic clothes are supposed to be the main source of airborne MPs, the release of synthetic textile fibers may be responsible for an important amount of indoor and outdoor human exposure [14,15,56]. It has been demonstrated that each item of clothing could be responsible for the release of about 1900 fibers per wash [6]. Moreover, in experiments using open petri dishes as blanks (i.e., negative control), background contamination in the laboratory was highlighted, probably due to airborne MPs originating from clothes [19,57,58]. With regard to concentrations of airborne MPs, little information is available. Some previous French studies revealed concentrations of 118 MPs m^2^/day in atmospheric fallout collected using a stainless-steel funnel connected to a 20 L glass bottle [14]. Subsequently, the same research group measured atmospheric fallout particles ranging from 53 to 110 m^2^/day, of which 29% were formed of MPs [15], whereas in the Chinese city of Dongguan, the concentration of fibrous and non-fibrous MPs from atmospheric deposition varied from 175 to 313 particles/m^2^/day [59]. In London, the MP deposition rate ranged from 575 to 1008 particles/m^2^/day [27].

This high variability is probably dependent on different climate conditions and seasonality, but also on different sampling methodology. Higher concentrations are thought to be found in indoor environments due to the presence of several sources, lower dilution volumes, and factors involved in the spreading of plastic particles. Concentrations in indoor air ranging from 3 to 15 particles/m^3^ have been found in a previous study [60]. Moreover, a recent French study also found plastic fiber concentrations in the indoor environment ranging between 0.4 and 59.5 particles/m^3^, whereas in the outdoor environment the concentration was 0.3 to 1.5 particles/m^3^ [54]. However, all of these values must be considered to be underestimates because the analytical techniques used to date have a high detection limit, and thus do not allow the complete identification of the most concerning fraction, i.e., the respirable fraction. Recently, a sophisticated analytical technique was used by Vianello et al. [61] associated with the classic Fourier transform infrared (FTIR) spectroscopy approach, called focal plane array (FPA) imaging analysis, to detect small MPs (lower limit 5.5 µm), reducing artefacts. As reported by the same authors, further improvement of the analytical sensitivity is necessary for the assessment of human exposure.

## 3. Factors Favoring Human Exposure to Airborne MP-NPs

Several factors influence the dispersion of any types of airborne particles and relative human exposure in both indoor and outdoor environments:vertical gradient, with higher concentrations of pollution near the ground and due to particle re-suspension;wind speed, with a decrease in pollution concentrations when the wind speed is higher;wind direction, parallel versus perpendicular to obstacles;sedimentation of particles >2.5 mm;temperature, with lower temperatures increasing nucleation and condensation of the particles resulting in lower atmospheric concentrations [62].

In outdoor environments, human exposure to MPs is largely influenced by meteorological and geographical factors. In particular, rainfall, wind, and local conditions greatly influence the atmospheric persistence of MPs and the subsequent fallout, in addition to particle sizes because larger particles sediment by gravity [14]. Therefore, the lightest MPs with the lowest density can be transported by the wind, resulting in more widespread environmental pollution [24,63]. Furthermore, the distribution of airborne particles in outdoor urban environments might be influenced by changes in the wind direction caused by urban topography (e.g., buildings and space between them), local meteorological conditions, and the presence of thermal circulation [64]. Subsequently, human exposure to low concentrations of airborne MPs due to particle dilution is thought to occur in outdoor environments [54]. However, in particular atmospheric conditions (e.g., poor ventilation), the dilution and removal of MPs may be reduced, resulting in a potentially higher-grade exposure.

The indoor presence of airborne MPs is dependent on room layout and ventilation, resulting in higher concentrations in poorly ventilated rooms. Seaton et al. [65] showed that airborne nanoparticles (Ø < 100 nm), of which nanoplastics are part, rapidly spread between compartments and remain floating in the air. For these reasons, and because people spend an average of 70–90% of their time indoors, the indoor exposure to airborne MPs appears to be highly relevant. Finally, indoor MPs may move to the outside air where they are dispersed in the atmosphere [54].

## 4. Occupational Exposure to MP-NPs

Obviously, the highest exposure to airborne MPs was more often derived from occupational environments rather than from staying at home, resulting in chronic exposure to these pollutants. In this regard, the possible onset of occupational diseases has been shown in certain categories of workers exposed to higher concentrations of airborne MPs. In particular, workers of two types of industries have been evaluated to study the consequences of airborne MP exposure: the synthetic textile industry, and the vinyl chloride (VC) and polyvinyl chloride (PVC) industries. Recently, a further source of airborne ultrafine plastic particles in the workplace has been represented by 3D printers, which use thermoplastics such as acrylonitrile butadiene styrene (ABS) and polylactic acid (PLA) [66].

In the synthetic textile industries, there is a high concentration of fibers composed of nylon, polyester, polyurethane, polyolefin, acrylic, and vinyl-type polymers. Most previous studies showed a link between the inhalation of these synthetic fibers and respiratory diseases [67,68]. Chronic exposure to irritants and persistent synthetic fibers has been evaluated as a cause of cancer, especially after 10–20 years [69,70]. Additionally, it has been shown that some natural fibers (e.g., hemp) may be responsible for dyspnoea more than synthetic fibers, probably due to their capacity to interfere with certain physiological functions [71]. However, evidence shows that cancer risk may be similar for natural and synthetic fibers [70]. A particular type of synthetic textile industry is the flock industry, in which velvet-like or fleeced fabrics are produced from pulverized or cut fibers, named flock, of 0.2–5.0 mm and often made of nylon, polyester, polyethylene, and polypropylene, applied to adhesive-coated materials [72,73]. From these synthetic fibers, inhalable particles are released during the cutting process [73,74] resulting in an interstitial lung disease named “flock worker’s lung” [74,75]. According to Washko et al. [75], the prevalence of systemic and respiratory symptoms in flock workers was 64.7%, and this value was related to hours worked per week. Furthermore, flock workers showed an increased odds ratio (OR 3.6; 95% CI: 1.07–12.02) for respiratory symptoms and a three-fold increase in the risk of developing lung cancer compared to non-exposed individuals [73]. In the pathogenesis of the disease, in addition to the toxicity caused by the fibers themselves, powders and finishing agents (e.g., tannic acid, potato starch), thermal degradation products (e.g., nitrogen dioxide), formaldehyde, metals, microorganisms, and even aflatoxins released by *Fusarium* sp. growing in adhesive barrels have been proposed [76,77]. Porter et al. [78] carried out an in vivo experiment using rats in which flock particles washed to remove contaminants were instilled intratracheally. The instillation induced symptoms similar to those observed in humans, and the washing medium was much less active [78].

In PVC industries, workers are exposed to VC and PVC (a white powder able to produce an inhalable dust). The pathogenic role of VC and PVC is shown by some studies, revealing the link between exposure to PVC dust and VC monomers and undifferentiated restrictive lung disease [79,80,81,82]. Toxicity may result from PVC dust, VC monomers, and thermal decomposition products [80]. PVC dust as a toxic agent is supported by multiple lines of evidence, such as the presence of PVC particles inside macrophages from human patients [83] and the onset of pathomorphological alterations in rat lungs instilled with PVC particles [84]. In addition, it has been shown that the toxicity of PVC dust may contribute to physical harm and the slow release of VC monomers from the particles to the adjacent lung tissue [81]. Moreover, in vitro experiments conducted on rat lung cells showed that using PVC particles containing additives produced a higher inflammatory potential than that triggered by additive-free PVC particles [82]. Finally, a carcinogenic role is linked both to PVC, responsible for an increased risk of lung cancer in workers, and VC, causing cancers in mice and rats [81].

## 5. Toxicity of Airborne MP-NPs

To clarify the toxicity of airborne MPs, several pathophysiological mechanisms have been found to cause inflammation, such as the weakening of clearance mechanisms determining dust overload, oxidative stress, cytotoxicity, and translocation [85,86,87,88]. All of these biological mechanisms probably concur and contribute in various degrees to the inflammatory response and the onset of cancerous lesions. In addition, toxicity from other chemical agents present on the particle surface may contribute to the inflammatory response.

After being inhaled, MPs reach the airways and are deposited dependent on the particle properties, patient characteristics, and lung anatomy; smaller particles with a lower density (e.g., PE) have a greater probability of reaching deep airways [89]. Particles of 5–30 μm deposit in the upper airways by impaction with the rhino-pharyngeal walls, whereas particles of 1–5 μm reach the small airways by sedimentation [89] and diffusion. The deposition of particles <1 μm occurs through Brownian motion [89,90].

After deposition, MP particles undergo clearance by several mechanisms, such as the mucous progression determined by ciliary movement, phagocytosis from alveolar macrophage, and migration through the lymphatic system [52]. However, these mechanisms are often insufficient to remove larger plastic fibers, and fibers sized up to 250 μm have been found in the deep lung [91,92].

Unlike the more efficient mechanism of mucociliary clearance, which acts exclusively in the upper respiratory tract allowing the removal of foreign material without internalization, direct cellular involvement occurs in phagocytosis, triggering the inflammatory cascade. Similar to pinocytosis, phagocytosis uses the energy-dependent endocytosis machinery and allows the internalization of particles of at least ~60 nm, through vesicles derived from the plasma membrane [30]. Almost all cells can internalize micro- and nanoparticles by pinocytosis (including alveolar epithelial cells), whereas phagocytosis is a feature of only specialized cells (i.e., professional phagocytes including macrophages, neutrophils, and monocytes). Both endocytosis processes involve actin polymerization by the stepwise use of GTPases [93].

In both endocytosis mechanisms, particle–cell surface interactions induce membrane invagination and vesicle formation containing foreign particles, which are internalized and subsequently fuse with lysosomes to form phagosomes with a diameter of 0.5–10 μm. Sometimes, phagosomes are destroyed, producing the known “lysosome-enhanced Trojan horse effect”, with intracellular toxicity and apoptosis due to acidification and the enzymolysis of lysosomes [94]. Moreover, as observed by us in A549 cells, the acidic compartment damage causes the release of intralysosomal redox-active iron due to the autophagocytic degradation of metalloproteins such as cytochromes. This endogenous source of metal causes the overproduction of reactive oxygen species (ROS), regardless of the metal contaminants in MP-NPs. In addition to the energy-dependent endocytic pathways, internalization can occur in all cell types by passive diffusion following the adhesive interactions of hydrophobic particles, such as MP-NPs, which can cross the membrane lipid bilayer. Due to membrane flexibility, the transport of these micro- and nanoparticles along the lipid bilayer does not cause a persistent hole, as observed in red blood cell membranes for which only energy-independent passive diffusion can take place [95].

The damage induced by the inhaled and internalized particles is especially due to the onset of inflammatory cellular responses, and it has been shown that even particles low in toxicity, which are poorly reactive, can result in disease in susceptible individuals, especially when clearance mechanisms are compromised [91,96,97,98,99]. The onset of inflammatory responses may be caused by dust overload, oxidative stress, and cytotoxicity. Dust overload can explain the effects of inert particles in the respiratory airways and consists of the accumulation of dust in the lung. It depends on the poor mobility of alveolar macrophages engulfed by a high particle amount or by particles with a high surface area, causing an intense release of chemotactic factors activating the migration of other cells, which is also favoured by increased vascular permeability [100]. The reduced clearance of particles by macrophages increases their lifetime in the interstitial spaces, which favors the cell–particle interaction and, consequently, their uptake, determining the onset of a chronic and intense inflammatory response [96,97]. According to the “overload paradigm”, chronic inflammation occurs when the pulmonary particle volume burden exceeds 6% of the macrophage volume in the lung. Once this dose threshold is reached, the mobility of alveolar macrophages and their ability to clear particles from the alveolar surface is retarded [101]. However, regardless of the burden, it has been hypothesized that the sharp-edged particles (non-spherical, abrasive MP-NPs) can cause lysosomal membrane destabilization, which contributes to the elevated inflammatory toxicity [102].

Triggering of the inflammation mechanism by persistent microfibers (longer than 5 µm) is characteristic. Unlike short microfibers, these are too long to be completely engulfed by alveolar macrophages causing the frustrated phagocytosis. This results in persistent inflammatory stimulation, leading to pulmonary fibrosis or even carcinogenesis [103].

Macrophages, in addition to phagocytizing this foreign matter and inducing the response of other immune cells, also release inflammasome, a multiprotein complex that controls the activation and maturation of the IL-1β cytokine [96]. IL-1β, together with IL-1R and IL-18, belongs to the IL-1 family of cytokines, which are important mediators of inflammatory responses. The most important activator of IL-1β and IL-18 processing and secretion is the NLRP3 inflammasome, whose activation is triggered by many different signals, including pathogen-associated molecular patterns (PAMPs) and danger-associated molecular patterns (DAMPs) [104].

Particles may determine oxidative stress by different mechanisms, such as the carriage of oxidizing species on their surface (e.g., metals) and the interaction of their high surface area with biological systems [105]. The resulting ROS overproduction, overcoming the cellular antioxidant systems, triggers cell signalling pathways, which are nuclear factor NF-kB-driven, of the inflammatory cascade with the release of cytokines (e.g., TNF-α, IL-β) and inflammatory mediators, resulting in inflammation and cytotoxicity [91,106,107]. Previous studies showed that different cultured cell lines, including rat and human alveolar epithelial cells, erythrocytes, and macrophages, are able to efficiently uptake PS and other particles [52,93,108,109,110]. In addition to cytotoxicity, the oxidative stress causes genotoxicity (i.e., DNA damage such as adducts and mutations) that may trigger the sequential carcinogenesis process. However, in cancer promotion, other mechanisms are involved, such as a particle’s direct action and the angiogenesis and mitogenesis processes promoted by pro-inflammatory mediators favoring the development and progression of malignant cells [91,96,106,111,112].

Therefore, chronic exposure to airborne MPs can result in cancer onset [70,91,113], as observed after 10–20 years in workers of the synthetic textile industry, with higher cancer incidence associated with the intensity and duration of the exposure [69,114]. In particular, PVC workers had an increased risk of lung cancer with an OR of 1.20 (95% CI: 1.1–1.3) related to exposure, age, and years at work [81]. Previous studies showed that the exposure to VC can induce mammary cancers in concentrations as low as 1–5 ppm [115]. Xia et al. [107] showed that PS microparticles are able to cause ROS production, while Bellmann et al. [116] found a clearance efficacy decrease following dust overload secondary to long-term exposure.

Dong et al. [117] found that PS microparticles induced oxidative stress and inflammatory responses with cytotoxic effects in human lung epithelial cells, and altered the epithelial layer in vitro. Recently, other studies have verified the toxicity of nanoplastics in lung and bronchial cells. Xu et al. [52] found that PS nanoparticles (Ø 25 and 70 nm) impaired viability, and induced cell cycle arrest and the up-regulation of nuclear factor NF-kB and some pro-inflammatory cytokines in the human alveolar epithelial line A549. By comparison, Lim et al. [118] noted that PS nanoparticles are cytotoxic only at high concentrations but, even at low concentrations, they induced metabolic alterations and stress in the endoplasmic reticulum in a human bronchial epithelial cell line.

As stated above, MPs from the erosion of agricultural land, use of fertilizers, dried sludge and products from waste water treatment, synthetic clothing fabrics, industrial emissions, and dust from the road and marine aerosols can be carried by the wind. Concerning the impact on human health, Pauly et al. [119] reported that 87% of examined human lungs contained plant fibers (for example cotton) and plastic fibers. The inhalation of these fibers is more likely for the smallest particles that can penetrate and persist in the lungs [33,60]. It has been shown that the outdoor areas with the greatest risks of exposure to airborne MP-NPs are urban roads and industrial areas [8]. Therefore, it is very likely that these fine and ultrafine plastics can be absorbed not only by eating contaminated food but also by talking or walking along a street [54]. The inhalation of MP-NPs could lead to breathing difficulties, and cytotoxic and inflammatory effects, in addition to autoimmune diseases in humans, because the lung has a very large alveolar surface of about 150 m^2^, with a very thin barrier of less than 1 µm, which could allow the nanoparticles to enter the bloodstream and move throughout the human body. Paget et al. [120] showed that the 50 nm-sized PS particles led to genotoxic and cytotoxic effects on lung epithelial cells and macrophages (Calu-3 and THP-1). Table summarizes the effects induced by different plastic materials tested in vitro on different cell lines (Table 1).

Inhaled particles, depending on the individual differences in metabolism and susceptibility, can cause:immediate bronchial reactions (asthma-like);diffuse interstitial fibrosis and granulomas with the inclusion of fibers (extrinsic allergic alveolitis, chronic pneumonia);inflammatory reaction and fibrotic changes in the bronchial and peribronchial tissue (chronic bronchitis);interalveolar septal lesions (pulmonary emphysema).

The cytotoxic effects implicated in the damage induced by exposure to MP-NPs are similar to those induced by exposure to other nanoparticles such as graphene-based nanoparticles (fullarene, and carbon nanotubes (CNTs)). Previous in vitro studies showed that CNTs are able to induce oxidative stress and mitochondrial impairment in the human alveolar cell line A549 exposed to multi-walled carbon nanotubes (MWCNTs) [48].

In addition to respiratory damage, inhaled particles may induce toxicity in other sites after blood translocation. Indeed, NP particles can cross the respiratory barrier and reach the blood, especially during inflammation when an increased endothelial and epithelial permeability is present. Intrinsic characteristics of the particle, such as hydrophobicity, size, and surface charge (the presence of positive charges induces higher translocation rates), affect the translocation [37,93]. Song et al. demonstrated the presence of particles in the intracellular space of pulmonary epithelial and mesothelial cells and in the thoracic effusion fluid after occupational exposure [121]. A similar situation has been described in subjects exposed to carbon nanoparticles by inhalation, which reached blood peak concentrations after 10–20 min [117,122]. Eyles et al. [123] conducted an in vivo study in rats treated with intranasal exposure to 1100 nm microplastics showing their presence in nasal associated lymphoid tissue (NALT), their passage from the lymphatic system to the bloodstream, and their accumulation in the spleen. A similar result was obtained in rats after gastrointestinal exposure to nanoplastics whose presence was detected in the systemic circulation, spleen, and liver. The ability of the MPs to undergo cellular uptake and cross biological barriers was assessed in some in vivo experiments. In particular, oxidative stress in hepatocytes was found in rats after the intravenous administration of 128 nm nanoplastics [124], and 240 nm PS particles used in a human placental perfusion model were able to cross the placental barrier [125]. Very recently, a study also ascertained the presence of microplastics in the placenta of pregnant women, causing concern in the scientific community and confirming what was previously only conceivable [126]. The translocation in the placental barrier can also occur after inhalation, as observed in rats after inhalation of rhodamine-labeled nanopolystyrene beads [127], highlighting the possibility of airborne MPs to determine toxicity in organs other than the respiratory system.

## 6. Contaminants Associated with MP-NPs

In addition to plastic monomers joined together by covalent bonds, plastic can also contain a wide variety of chemical products, used as additives. Unlike monomers, additives are not covalently bonded to the polymer, so they can be released into the surrounding environment [128].

The molecular weight of the additives can influence the release of MPs in the various environmental matrices (air, water, and soil). It is estimated that the molecular weight of substances used as additives in plastics ranges between 200 and 2000 g/mol. The phenomenon occurring when chemicals in plastic move from the surface of the material to the substrate in contact with the same material is known as “migration”. The migration rate of organic chemicals depends on their steric hindrance and compounds with low molecular weight, typically monomers and residual solvents, migrate at a faster rate than molecules with a high molecular weight [129].

Additives are organic and inorganic compounds added to plastics to give certain chemical-physical characteristics. They can be grouped into different categories:functional additives (stabilizers, flame-retardants, plasticizers, etc.);dyes;inert fillers (talc, kaolin, clay, calcium carbonate);fillers (for example glass fibers).

Flame-retardants are stabilizing additives used to slow or block the spread of flames in the event of a fire. Therefore, they allow the combustion process to be delayed by cooling the combustible material through a physical or chemical action. Among the used compounds, there are the organic halides, containing chlorine and bromine. Specifically, brominated flame-retardants (BFRs) are added to a wide category of products, such as electronic devices, textiles, and upholstery. BFRs can contaminate the environment and are therefore ubiquitous in different settings such as air, dust, soil, and sediment [102,130], in addition to living organisms (including humans) [131]. Due to this environmental contamination and the evidence of their toxicity, BFRs have been of a great concern. Therefore, some BFRs, such as polybrominated diphenyl ethers (PBDEs), have been listed as persistent organic pollutants (POPs) by the Stockholm Convention and are subject to restrictions on their production and use in a number of countries [132]. However, even for the new BFRs (NBFRs), which are now considered a safer alternative to prohibited compounds, the behavior is not yet fully understood concerning either toxicity or long-term action [133]. These compounds have chemical-physical structures and characteristics that are very similar to those of legacy flame-retardants due to their aromatic structure, the high degree of halogenation, and their low solubility in water. Therefore, similar to the prohibited compounds, it is likely that they are toxic, able to bioaccumulate in organisms, and persistent in the environment. Traces of NBFRs have already been found in the Arctic region distributed in both soil and sediments [134].

Plasticizers are the most commonly used additives and are added to increase plasticity and flexibility. Their molecules are much smaller than polymers and can enter homogeneously between the polymer macromolecules. Examples of plasticizers are phthalates, esters of phthalic acid, and bisphenol A, which are low volatility substances that are very soluble in oils, but less so in water, and are colorless and odorless. Many studies have shown that phthalates and bisphenol A are responsible for negative effects on human health, especially in children, because they can act as endocrine disruptors and carcinogens and cause respiratory diseases [135]. European countries, and others, have introduced restrictions regarding the use of children’s toys containing phthalates, the concentration of which must not exceed 0.1% [136]. Finally, the majority of the dyes used are insoluble pigments divided into organic and inorganic categories. Organic pigments give plastics a bright color and a high resistance to light and atmospheric agents. For these reasons, they are used more frequently than inorganic pigments. Indeed, inorganic pigments have a lower coloring power, meaning that the color is not very bright, and they can also contain heavy metals [134].

Moreover, in addition to the compounds artificially added during the production process, MPs can absorb several types of contaminants from the environment due to their large surface area [137]. The damage processes of MPs linked to the environmental exposure determine their changes and modify their physicochemical characteristics. For example, although it has been shown that surface oxidation may increase their affinity for metals, the affinity for hydrophobic compounds decreases [5]. In other words, old plastics modified by environmental exposure can have different adsorption affinities to their virgin plastics. In marine MPs, persistent organic pollutants (POPs) have also been detected [138]. However, the same fate can happen in MPs suspended in the atmosphere. Indeed, some POPs formed during the combustion process, such as polychlorinated dibenzodioxin and dibenzofurans, have been already detected in atmospheric PM [139]. In the atmosphere, MPs could be exposed to POPs but the adsorption is dependent on the time at which plastic particles are suspended. Finally, many studies have shown the presence of heavy metals in MPs from sediments, soil, the marine environment, and organisms [140]. However, little is still known about the absorption of metals by airborne MPs, unlike that which has already been shown for PM and other types of particles [46,141,142,143]. As for other air pollutants, identification of the role played by each agent in the pathophysiology of the damage is difficult and toxicity could also result from complex interactions between the mixtures of pollutants. Further research is necessary to clarify the role of each agent, especially in occupational exposure.

## 7. Conclusions

The scientific evidence highlights the role of the air compartment as an important source of human exposure to MPs. Studies showed that airborne MPs are particularly made up of fibers ranging between 200 and 600 μm [14,15]. Because fibers up to 250 μm have been found in human lungs [119], it is possible that vulnerable individuals are at risk to develop respiratory diseases following their inhalation. Considering the fiber concentrations present in the indoor environment [60,92], and that the estimated human respiratory total volume is of 6 L min^−1^ under normal conditions, i.e., at rest or with low physical activity, it has been hypothesized that a person could be exposed to 26–130 airborne MPs per day. This condition could represent a risk for human health, especially in susceptible individuals (e.g., children and elderly), for a number of reasons, including the difficult removal from the airways due to the polymeric persistent structure and fibrous shape, and the pro-oxidant and pro-inflammatory effects, among others. Moreover, MPs often contain harmful additives such as plasticizers, which act as endocrine disruptors and are easily subject to migration [134,135,136,137]. Finally, airborne MPs could also adsorb dangerous pollutants, such as POPs, during their long stay in the environment, delivering them in high concentrations to human lungs upon release from their surface.

Because exposure to these levels lasts a lifetime (24/24 h), regardless of our behaviors with even greater peaks in certain conditions, we believe that inhalation is far more prevalent than ingestion, which is perceived by the population as being more dangerous. For all of these reasons, it is of primary importance for public health to increase the interest in these emerging air pollutants together with other classic air contaminants, such as PM combustion-produced and gaseous pollutants. In particular, the correct use and management of the plastic items and their recycling must be stressed. This aim can only be reached by grasping the interest of the entire community. Previous studies have investigated the opinions of young people regarding environmental pollution, showing that the interest in this issue and the derived pathologies is very high. Considering the importance of the environment on specific health problems, respiratory diseases were the primary concern taken into consideration, followed by tumors, infectious diseases, congenital malformations, heart diseases, and neurological disorders [144]. This shows the importance of increasing the knowledge of these specific air pollutants in addition to classical pollutants, and to promote a green culture to increase the responsibility of each individual and the entire community.

## Figures and Tables

**Figure 1 ijerph-18-02997-f001:**
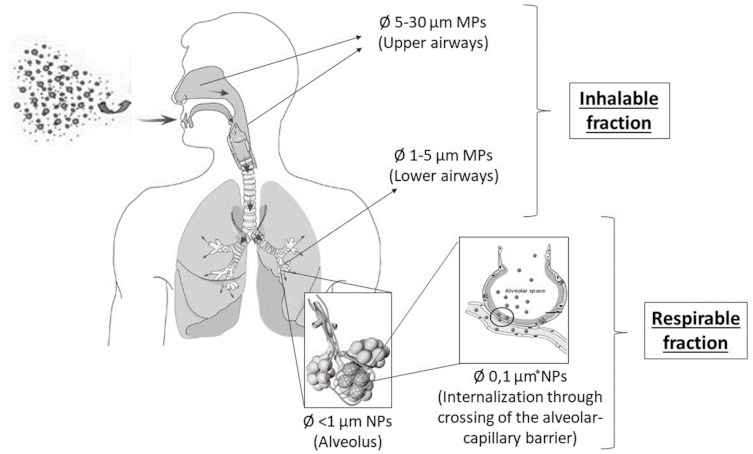
Size-selective inhalation of airborne microplastics involves specific regions of the respiratory tract: inhalable fraction (upper and lower airways) and respirable fraction (deep airway).

**Table 1 ijerph-18-02997-t001:** In vitro studies of plastic material effects: oxidative stress, mitochondrial damage, cytotoxicity, and inflammation in different cell lines.

Authors	Plastic Material	Cell Lines	Time of Exposure	Effects
Wu et al. [40]	PS	Caco-2	12 h	Significant increases in intracellular ROS levels, mitochondrial depolarization and plasma membrane ABC transporters inhibition.
Xu et al. [52]	PS	A549	24 h	PS-NPs significantly affected the cell viability, caused cell cycle S phase arrest, activated inflammatory gene transcription, and changed the expression of proteins associated with cell cycle and apoptosis. PS-NPs induced significant up-regulation of pro-inflammatory cytokines such as IL-8, NF-κB, and TNF-α, as well as pro-apoptotic proteins such as DR5, caspase-3, caspase-8, caspase-9, and cytochrome c
Hoelting et al. [85]	PE-NPs	hESCs	84 h-18 days	Increased cytotoxicity and oxidative stress (dose-dependent). 18-day exposure: PE-NP accumulation (≥22.6 mg/L). Altered gene expression (22.5 mg/L) and increased cytotoxicity (≥180 μg/mL).
Murali et al. [86]	PS-PEG and PS-COOH	Primary mouse astrocytes, neurons, microglia and brainvascular endothelial cells	24 h	Decreased mitochondrial activity and cell viability.
Schirinzi et al. [87]	PE,PS	T98G, HeLa	24–48 h	PE caused ROS generation on T98G only. In both cell cultures PS caused ROS generation. Oxidative stress can explains the toxicity observed for both T98G and HeLa cells.
Poma et al. [88]	PS	Hs-27	24 h	PS NPs stimulated ROS production and induced genotoxic stress and DNA damage as measured with the cytokinesis-block micronucleus (CBMN) assay
Prietl et al. [97]	PS-COOH	THP-1, DMBM-2	4–24 h	PS particles stimulated IL-6 and IL-8 secretion in monocytes and macrophages, chemotaxis towards a chemotactic stimulus of monocytes and phagocytosis of bacteria by macrophages and provoked an oxidative burst of granulocytes.
Xia et al. [107]	PS	RAW 264.7	//	Cationic PS nanospheres induced cellular ROS production, GSH depletion, mitochondrial damage and cell death without inflammation.
Dong et al. [117]	PS	BEAS-2B	24–48 h	After 24 h of exposure, a significant reduction in cell viability and ROS accumulation were observed only in the PS-MP-treated BEAS-2B cells at a concentration of 1000 μg/cm^2^.
Lim et al. [118]	PS	BEAS-2B	//	PS exposure showed autophagic- and endoplasmic reticulum (ER) stress-related metabolic changes such as increased in amino acids and tricarboxylic acid cycle (TCA) intermediate metabolites.
Paget et al. [120]	PS-NF, PS-COOH, PS-NH_2_	THP-1, Calu-3	24–48 h	PS-NF and PS-COOH did not induce significant responses compared to non-exposed cells. Strong effects on cell viability were caused by PS-NH_2_ which induced a dose-dependent Cell Index (CI) decreases for all concentrations.

PS: polystyrene; Caco-2: human colorectal adenocarcinoma cell line; A549: human lung carcinoma cell line; PE-NPs: polyethylene nanoparticles; hESCs: human embryonic stem cells; PS-PEG: polystyrene PEGylated; PS-COOH: polystyrene carboxylated; PE: polyethylene; T98G: glioblastoma cell line; HeLa: human cervix epitheloid carcinoma cell line; Hs-27: human foreskin fibroblast cell line; THP-1: human monocytic cell line; DMBM-2: murine macrophages cell line; RAW 264.7: murine macrophage cell line; BEAS-2B: human normal bronchial epithelium cell line; PS-NF: polystyrene nonfunctionalized; PS-NH_2_: polystyrene aminated; Calu-3: human lung cancer cell line.

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
