# Peer review of "Newly Emerging Airborne Pollutants: Current Knowledge of Health Impact of Micro and Nanoplastics"

_ijerph, 2021, doi:10.3390/ijerph18062997_

Round 1

Reviewer 1 Report

This review article is an important and interesting topic. It also surveys SOURCES AND CONCENTRATION, several possible FACTORS, EXPOSURE, TOXICITY, and CONTAMINANTS. The review is quite comprehensive. However, I think more studies about human health effects or risks can be addressed. Furthermore, I suggest some figures and tables can be included in this paper. It will help readers to easily understand this topic. Also, the abstract should include some quantitative conclusions.

Author Response

This review article is an important and interesting topic. It also surveys SOURCES AND CONCENTRATION, several possible FACTORS, EXPOSURE, TOXICITY, and CONTAMINANTS. The review is quite comprehensive. However, I think more studies about human health effects or risks can be addressed. Furthermore, I suggest some figures and tables can be included in this paper. It will help readers to easily understand this topic. Also, the abstract should include some quantitative conclusions.

We thank the reviewer for the positive opinion expressed and, accepting the suggestions, we have added a table where the results obtained from in vitro studies on human and other mammalian cell lines are summarized.

We addedd a Figure which schematizes the size-dependent pathways of airborne micro and nano plastic particles.

Further studies about human health effects or risks were addedd (Hoelting et al., 2013; Prietl et al., 2014; Murali et al., 2015; Schirinzi et al., 2017; Poma et al., 2019; Ragusa et al., 2021; Fournier et al., 2020). In particular in the study of Ragusa et al. (2021) the presence of MPs has ascertained in placenta of pregnant women.

Reviewer 2 Report

This study summarized the toxicity and potential health impact of airborne MPs (microplastics) and NPs (nanoplastics) from reviewing related studies. Though MPs was found in human lung with at 250 mm, they are still hard to be inhalable and travel further to reach human cell. NPs with tiny size are still the more dangerous pollutants which were less mentioned here, most of reviewed papers regarding toxicity using commercial NPs not real world airborne NPs, therefore, it is insufficient to include NPs in this report. Here are some specific comments

  1. Line 48, 70: The classification of MPs was not clear. MPs was defined as the pieces smaller than 5 mm, and classified into medium-sized MPs (1.01-4.75 mm) and small MPs (<1 mm) what are MPs with size (4.75-5mm)?
  2. Line 104-112: The sources of MPs was mentioned but not NPs. Whether 3D printing with thermoplastic (the most concerned topic related to airborne NPs emission) can be considered as the sources of airborne NPs? What is the characteristics of MPs for each sources?
  3. Line 139-169: The factors were proved effecting on the distribution and dispersion of airborne MPs, however, the authors claimed those factors influenced on human exposure are insufficient.

Author Response

This study summarized the toxicity and potential health impact of airborne MPs (microplastics) and NPs (nanoplastics) from reviewing related studies. Though MPs was found in human lung with at 250 mm, they are still hard to be inhalable and travel further to reach human cell. NPs with tiny size are still the more dangerous pollutants which were less mentioned here, most of reviewed papers regarding toxicity using commercial NPs not real world airborne NPs, therefore, it is insufficient to include NPs in this report. Here are some specific comments

The reviewer is certainly right, unfortunately very little is known about the biological effects and concentrations of nanoplastics present in the various environmental matrices. As reported in our paper (lines 99-100), we do not even know the exposure doses as “a standardised method for their analysis is still missing”. Doubtless, these are also inhaled and, to bypass this lack of knowledge, in our paper we focused biological effects of other nanosized particles (carbon- and metal-based nanomaterial). At least in part considering the lower reactivity, these effects could occur after inhalation of nanoplastics, as in vitro studies using commercial NPs seem to confirm. In comparison to NPs subjected to environmental aging, virgin NPs have a lower ability to interact with biological systems, making the damage observed in vitro plausibly underestimated. Even if it could be considered premature to discuss the health impact of nanoplastics, we consider it crucial to underline what is already known about the toxicity of nanosized particles to emphasize the potential damage of nanoplastics. Like us, a very recent review (Prüst M, et al. The plastic brain: neurotoxicity of micro- and nanoplastics. Part Fibre Toxicol.2020.) extrapolates the neurotoxicity of nanoplastics from studies carried out on other nanoparticles with similar characteristics (poor reactivity) such as metal(oxide) nanoparticles of gold and titanium.

  1. Line 48, 70: The classification of MPs was not clear. MPs was defined as the pieces smaller than 5 mm, and classified into medium-sized MPs (1.01-4.75 mm) and small MPs (<1 mm) what are MPs with size (4.75-5mm)?

We changed 4.75 with 5mm

  1. Line 104-112: The sources of MPs was mentioned but not NPs. Whether 3D printing with thermoplastic (the most concerned topic related to airborne NPs emission) can be considered as the sources of airborne NPs? What is the characteristics of MPs for each sources?

Released into the environment by 3D printers, nanoplastics share with microplastics With the exception of nanoplastics the same sources and their environmental presence is due to the further fragmentation, wear and degradation of plastic materials (lines 52-53).

  1. Line 139-169: The factors were proved effecting on the distribution and dispersion of airborne MPs, however, the authors claimed those factors influenced on human exposure are insufficient.

We carefully reread this part of our paper (Lines 139-169) and we could not find a sentence in which “we claimed those factors influenced on human exposure are insufficient”.

Reviewer 3 Report

As a reviewer I have the following remarks.

  1. Abstract: “PVC”? – please use full spelling and (PVC).
  2. Line 256. ROS is used but it’s defined later Line 297.
  3. Line 311 – I am guessing OR =odds ratio.
  4. The paper is well written and presented.

I have some additional questions to them:

1.Line 66: “Despite fragments being considered the most abundant form, fibres appear to be far more represented and accounted for 67% of all atmospheric MPs in Shanghai [26] and 92% in London [27].” – do you known the reason of fibres present in large cities?

2.Do we see the differences between urban vs. rural areas (as there are not organic plastic) with respect to MP?

3.In environmental epidemiology we consider fine particulate matter (PM2.5) levels. According to this presentation, probably some % of their mass belongs to MP?
Dou you think that a mixture metals and MP is more toxic?

Thank you

Author Response

As a reviewer I have the following remarks.

  1. Abstract: “PVC”? – please use full spelling and (PVC).

We added full spelling

  1. Line 256. ROS is used but it’s defined later Line 297.

We changed, shifting the definition in line 256

  1. Line 311 – I am guessing OR =odds ratio.

In lines 193-194 we used the terms odds ratio with the acronym OR in brackets, so subsequently, we used only OR

  1. The paper is well written and presented.

Thanks for the judgment

I have some additional questions to them:

1.Line 66: “Despite fragments being considered the most abundant form, fibres appear to be far more represented and accounted for 67% of all atmospheric MPs in Shanghai [26] and 92% in London [27].” – do you known the reason of fibres present in large cities?

The environmental monitoring of MP-NPs is not yet standardized, so the differences could also partly be due to the different methods of analysis. Furthermore, the composition is also subject to seasonal variations and the authors of the two studies in Shanghai and London do not tell us at what time of year they detected such a high presence of fibers.

2.Do we see the differences between urban vs. rural areas (as there are not organic plastic) with respect to MP?

A difference between urban and rural air is conceivable due to lower sources (vehicular traffic, tire wear, population density and consequent lower release of synthetic fibers from clothing). However, in rural environments should not be underestimated the use of fertilizers, the plastic used for covering greenhouses, some agricultural practices, the spreading of sewage sludge etc.

3.In environmental epidemiology we consider fine particulate matter (PM2.5) levels. According to this presentation, probably some % of their mass belongs to MP?
Dou you think that a mixture metals and MP is more toxic?

As suggested by the reviewer, certainly a fraction of the PM (to our knowledge still never quantified) is made up of plastic polymers. It will be interesting to evaluate the health impact of this complex mixture (i.e. PM) in which organic (Polycyclic Aromatic Hydrocarbon -PAH, dioxin, dibenzofurans, MP-NPs etc.) and inorganic (metal) pollutants are present. In our paper (lines 430-433) we underlined that surface oxidation of MP-NPs may increase their affinity for metals. This will make the metals more bioavailable, favoring their transmembrane passage.

Reviewer 4 Report

The manuscript is timely and addresses an important topic, i.e., the potential toxicity of inhaled micro and nano plastics. The manuscript covers quite well the aspects of production and environmental fate of plastics and the basic mechanisms of their toxicity. However, there are points that shold be clarified before considering the manuscript for potential publication, such as:

a) in page 7, authors list a series of pulmonary responses or diseases that may occur due to microplastic inhalation. There are some inappropiate definitions. Interalveolar septal destruction is the basic event in pulmonary emphysema and not pneumothorax, which is the scape fo air to the interpleural space due to the rupture of pleural membrane.

In addition, in my opinion, should be less emphatic in their statements on the possible adverse effects of plastics  in their conclusion. They are right in addresseing the potential toxicity, but, up to now, there are not enough evidences to establish certainty, Their review clearly will motivate several groups to conduct studies to explore the multiple potential adverse effects of plastics.

Author Response

The manuscript is timely and addresses an important topic, i.e., the potential toxicity of inhaled micro and nano plastics. The manuscript covers quite well the aspects of production and environmental fate of plastics and the basic mechanisms of their toxicity. However, there are points that shold be clarified before considering the manuscript for potential publication, such as:

  1. a) in page 7, authors list a series of pulmonary responses or diseases that may occur due to microplastic inhalation. There are some inappropiate definitions. Interalveolar septal destruction is the basic event in pulmonary emphysema and not pneumothorax, which is the scape fo air to the interpleural space due to the rupture of pleural membrane.

Apologizing for the gross mistake, we corrected changing pneumothorax with pulmonary emphysema

In addition, in my opinion, should be less emphatic in their statements on the possible adverse effects of plastics in their conclusion. They are right in addresseing the potential toxicity, but, up to now, there are not enough evidences to establish certainty, Their review clearly will motivate several groups to conduct studies to explore the multiple potential adverse effects of plastics.

As hypothesized by the reviewer, human MP-NPs exposure is a paramount topic for public health and our group is also performing studies to assess the pathogenic mechanism of MP-NPs by comparing the virgin particles to those subjected to environmental aging.

Round 2

Reviewer 2 Report

This paper now much better than that of previous version, I agree to publish this paper.